# Online Variational Approximations to non-Exponential Family Change Point Models: With Application to Radar Tracking

**Ryan Turner**
Northrop Grumman Corp.
ryan.turner@ngc.com

**Steven Bottone**
Northrop Grumman Corp.
steven.bottone@ngc.com

**Clay Stanek**
Northrop Grumman Corp.
clay.stanek@ngc.com

## Abstract

The Bayesian online change point detection (BOCPD) algorithm provides an efficient way to do exact inference when the parameters of an underlying model may suddenly change over time. BOCPD requires computation of the underlying model's posterior predictives, which can only be computed online in $O(1)$ time and memory for exponential family models. We develop variational approximations to the posterior on change point times (formulated as run lengths) for efficient inference when the underlying model is not in the exponential family, and does not have tractable posterior predictive distributions. In doing so, we develop improvements to online variational inference. We apply our methodology to a tracking problem using radar data with a signal-to-noise feature that is Rice distributed. We also develop a variational method for inferring the parameters of the (non-exponential family) Rice distribution.

Change point detection has been applied to many applications [5; 7]. In recent years there have been great improvements to the Bayesian approaches via the Bayesian online change point detection algorithm (BOCPD) [1; 23; 27]. Likewise, the radar tracking community has been improving in its use of *feature-aided tracking* [10]: methods that use auxiliary information from radar returns such as signal-to-noise ratio (SNR), which depend on radar cross sections (RCS) [21]. Older systems would often filter only noisy position (and perhaps Doppler) measurements while newer systems use more information to improve performance. We use BOCPD for modeling the RCS feature. Whereas BOCPD inference could be done exactly when finding change points in conjugate exponential family models the physics of RCS measurements often causes them to be distributed in non-exponential family ways, often following a Rice distribution. To do inference efficiently we call upon variational Bayes (VB) to find approximate posterior (predictive) distributions. Furthermore, the nature of both BOCPD and tracking require the use of online updating. We improve upon the existing and limited approaches to online VB [24; 13]. This paper produces contributions to, and builds upon background from, *three* independent areas: change point detection, variational Bayes, and radar tracking.

Although the emphasis in machine learning is on filtering, a substantial part of tracking with radar data involves *data association*, illustrated in Figure 1. Observations of radar returns contain measurements from multiple objects (targets) in the sky. If we knew which radar return corresponded to which target we would be presented with $N_T \in \mathbb{N}_0$ independent filtering problems; Kalman filters [14] (or their nonlinear extensions) are applied to "average out" the *kinematic errors* in the measurements (typically positions) using the measurements associated with each target. The *data association problem* is to determine which measurement goes to which track. In the classical setup, once a particular measurement is associated with a certain target, that measurement is plugged into the filter for that target as if we knew with certainty it was the correct assignment. The association algorithms, in effect, find the maximum a posteriori (MAP) estimate on the measurement-to-track association. However, approaches such as the joint probabilistic data association (JPDA) filter [2] and the probability hypothesis density (PHD) filter [16] have deviated from this.

To find the MAP estimate a log likelihood of the data under each possible *assignment vector* **a** must be computed. These are then used to construct cost matrices that reduce the assignment problem to a particular kind of optimization problem (the details of which are beyond the scope of this paper). The motivation behind feature-aided tracking is that additional features increase the probability that the MAP measurement-to-track assignment is correct. Based on physical arguments the RCS feature (SNR) is often Rice distributed [21, Ch. 3]; although, in certain situations RCS is exponential or gamma distributed [26]. The parameters of the RCS distribution are determined by factors such as the shape of the aircraft facing the radar sensor. Given that different aircraft have different RCS characteristics, if one attempts to create a continuous track estimating the path of an aircraft, RCS features may help distinguish one aircraft from another if they cross paths or come near one another, for example. RCS also helps distinguish genuine aircraft returns from *clutter*: a flock of birds or random electrical noise, for example. However, the parameters of the RCS distributions may also change for the same aircraft due to a change in angle or ground conditions. These must be taken into account for accurate association. Providing good predictions in light of a possible sudden change in the parameters of a time series is "right up the alley" of BOCPD and change point methods.

The original BOCPD papers [1; 11] studied sudden changes in the parameters of exponential family models for time series. In this paper, we expand the set of applications of BOCPD to radar SNR data which often has the same change point structure found in other applications, and requires online predictions. The BOCPD model is highly modular in that it looks for changes in the parameters of any underlying process model (UPM). The UPM merely needs to provide posterior predictive probabilities, the UPM can otherwise be a "black box." The BOCPD queries the UPM for a prediction of the next data point under each possible *run length*, the number of points since the last change point. If (and only if by Hipp [12]) the UPM is exponential family (with a conjugate prior) the posterior is computed by accumulating the sufficient statistics since the last potential change point. This allows for $\mathcal{O}(1)$ UPM updates in both computation and memory as the run length increases. We motivate the use of VB for implementing UPMs when the data within a *regime* is believed to follow a distribution that is not exponential family. The methods presented in this paper can be used to find variational run length posteriors for general non-exponential family UPMs in addition to the Rice distribution. Additionally, the methods for improving online updating in VB (Section 2.2) are applicable in areas outside of change point detection.

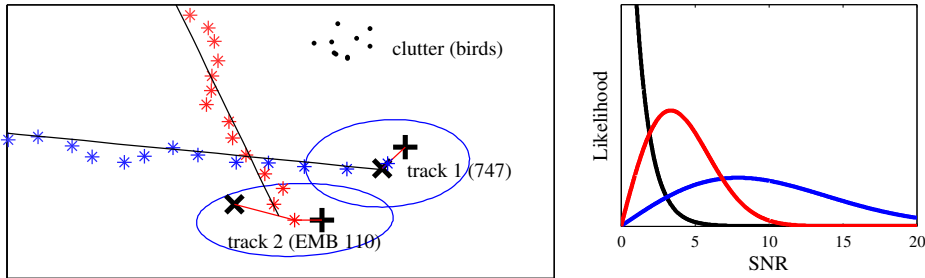

Figure 1: Illustrative example of a tracking scenario: The black lines (−) show the true tracks while the red stars (∗) show the state estimates over time for track 2 and the blue stars for track 1. The 95% credible regions on the states are shown as blue ellipses. The current (+) and previous (×) measurements are connected to their *associated* tracks via red lines. The clutter measurements (birds in this case) are shown with black dots (·). The distributions on the SNR (RCS) for each track (blue and red) and the clutter (black) are shown on the right.

To our knowledge this paper is the first to demonstrate how to compute Bayesian posterior distributions on the parameters of a Rice distribution; the closest work would be Lauwers et al. [15], which computes a MAP estimate. Other novel factors of this paper include: demonstrating the usefulness (and advantages over existing techniques) of change point detection for RCS estimation and tracking; and applying variational inference for UPMs where analytic posterior predictives are not possible. This paper provides four main technical contributions: 1) VB inference for inferring the parameters of a Rice distribution. 2) General improvements to online VB (which is then applied to updating the UPM in BOCPD). 3) Derive a VB approximation to the run length posterior when the UPM posterior predictive is intractable. 4) Handle censored measurements (particularly for a Rice distribution) in VB. This is key for processing missed detections in data association.

# 1   Background

In this section we briefly review the three areas of background: BOCPD, VB, and tracking.

## 1.1   Bayesian Online Change Point Detection

We briefly summarize the model setup and notation for the BOCPD algorithm; see [27, Ch. 5] for a detailed description. We assume we have a time series with $n$ observations so far $y_1, \ldots, y_n \in \mathcal{Y}$. In effect, BOCPD performs message passing to do online inference on the run length $r_n \in 0{:}n-1$, the number of observations since the last change point. Given an *underlying predictive model* (UPM) and a *hazard function* $h$, we can compute an exact posterior over the run length $r_n$. Conditional on a run length, the UPM produces a sequential prediction on the next data point using all the data since the last change point: $p(y_n|\mathbf{y}_{(r)}, \Theta_m)$ where $(r) := (n-r){:}(n-1)$. The UPM is a simpler model where the parameters $\theta$ change at every change point and are modeled as being sampled from a prior with hyper-parameters $\Theta_m$. The canonical example of a UPM would be a Gaussian whose mean and variance change at every change point. The online updates are summarized as:

$$\mathrm{msg}_n := p(r_n, \mathbf{y}_{1:n}) = \sum_{r_{n-1}} \underbrace{P(r_n|r_{n-1})}_{\text{hazard}} \underbrace{p(y_n|r_{n-1}, \mathbf{y}_{(r)})}_{\text{UPM}} \underbrace{p(r_{n-1}, \mathbf{y}_{1:n-1})}_{\text{msg}_{n-1}} . \tag{1}$$

Unless $r_n = 0$, the sum in (1) only contains one term since the only possibility is that $r_{n-1} = r_n - 1$. The indexing convention is such that if $r_n = 0$ then $y_{n+1}$ is the first observation sampled from the new parameters $\theta$. The marginal posterior predictive on the next data point is easily calculated as:

$$p(y_{n+1}|\mathbf{y}_{1:n}) = \sum_{r_n} p(y_{n+1}|\mathbf{y}_{(r)}) P(r_n|\mathbf{y}_{1:n}) . \tag{2}$$

Thus, the predictions from BOCPD fully integrate out any uncertainty in $\theta$. The message updates (1) perform exact inference under a model where the number of change points is not known a priori.

**BOCPD RCS Model**   We show the Rice UPM as an example as it is required for our application. The data within a regime are assumed to be iid Rice observations, with a normal-gamma prior:

$$y_n \sim \mathrm{Rice}(\nu, \sigma) , \quad \nu \sim \mathcal{N}(\mu_0, \sigma^2/\lambda_0) , \quad \sigma^{-2} =: \tau \sim \mathrm{Gamma}(\alpha_0, \beta_0) \tag{3}$$

$$\implies p(y_n|\nu, \sigma) = y_n \tau \exp(-\tau(y_n^2 + \nu^2)/2) I_0(y_n \nu \tau) \mathbb{I}\{y_n \geq 0\} \tag{4}$$

where $I_0(\cdot)$ is a modified Bessel function of order zero, which is what excludes the Rice distribution from the exponential family. Although the normal-gamma is not conjugate to a Rice it will enable us to use the VB-EM algorithm. The UPM parameters are the Rice shape[1] $\nu \in \mathbb{R}$ and scale $\sigma \in \mathbb{R}^+$, $\theta := \{\nu, \sigma\}$, and the hyper-parameters are the normal-gamma parameters $\Theta_m := \{\mu_0, \lambda_0, \alpha_0, \beta_0\}$.

Every change point results in a new value for $\nu$ and $\sigma$ being sampled. A posterior on $\theta$ is maintained for each run length, i.e. every possible starting point for the current regime, and is updated at each new data point. Therefore, BOCPD maintains $n$ distinct posteriors on $\theta$, and although this can be reduced with *pruning*, it necessitates posterior updates on $\theta$ that are computationally efficient.

Note that the run length updates in (1) require the UPM to provide predictive log likelihoods at all sample sizes $r_n$ (including zero). Therefore, UPM implementations using such approximations as plug-in MLE predictions will not work very well. The MLE may not even be defined for run lengths smaller than the number of UPM parameters $|\theta|$. For a Rice UPM, the efficient $\mathcal{O}(1)$ updating in exponential family models by using a conjugate prior and accumulating sufficient statistics is not possible. This motivates the use of VB methods for approximating the UPM predictions.

## 1.2   Variational Bayes

We follow the framework of VB where when computation of the exact posterior distribution $p(\theta|\mathbf{y}_{1:n})$ is intractable it is often possible to create a variational approximation $q(\theta)$ that is locally optimal in terms of the Kullback-Leibler (KL) divergence $\mathrm{KL}(q\|p)$ while constraining $q$ to be in a certain family of distributions $\mathcal{Q}$. In general this is done by optimizing a lower bound $\mathcal{L}(q)$ on the evidence $\log p(\mathbf{y}_{1:n})$, using either gradient based methods or standard fixed point equations.

**The VB-EM Algorithm** In many cases, such as the Rice UPM, the derivation of the VB fixed point equations can be simplified by applying the VB-EM algorithm [3]. VB-EM is applicable to models that are conjugate-exponential (CE) after being augmented with latent variables $\mathbf{x}_{1:n}$. A model is CE if: 1) The complete data likelihood $p(\mathbf{x}_{1:n}, \mathbf{y}_{1:n}|\theta)$ is an exponential family distribution; and 2) the prior $p(\theta)$ is a conjugate prior for the complete data likelihood $p(\mathbf{x}_{1:n}, \mathbf{y}_{1:n}|\theta)$. We only have to constrain the posterior $q(\theta, \mathbf{x}_{1:n}) = q(\theta)q(\mathbf{x}_{1:n})$ to factorize between the latent variables and the parameters; we do not constrain the posterior to be of any particular parametric form. Requiring the complete likelihood to be CE is a much weaker condition than requiring the marginal on the observed data $p(\mathbf{y}_{1:n}|\theta)$ to be CE. Consider a mixture of Gaussians: the model becomes CE when augmented with latent variables (class labels). This is also the case for the Rice distribution (Section 2.1).

Like the ordinary EM algorithm [9] the VB-EM algorithm alternates between two steps: 1) Find the posterior of the latent variables treating the expected natural parameters $\bar{\eta} := \mathbb{E}_{q(\theta)}[\eta]$ as correct: $q(x_i) \leftarrow p(x_i|y_i, \eta = \bar{\eta})$. 2) Find the posterior of the parameters using the expected sufficient statistics $\bar{\mathcal{S}} := \mathbb{E}_{q(\mathbf{x}_{1:n})}[\mathcal{S}(\mathbf{x}_{1:n}, \mathbf{y}_{1:n})]$ as if they were the sufficient statistics for the complete data set: $q(\theta) \leftarrow p(\theta|\mathcal{S}(\mathbf{x}_{1:n}, \mathbf{y}_{1:n}) = \bar{\mathcal{S}})$. The posterior will be of the same exponential family as the prior.

### 1.3 Tracking

In this section we review data association, which along with filtering constitutes tracking. In data association we estimate the *association vectors* $\mathbf{a}$ which map measurements to tracks. At each time step, $n \in \mathbb{N}_1$, we observe $N_Z(n) \in \mathbb{N}_0$ measurements, $Z_n = \{\mathbf{z}_{i,n}\}_{i=1}^{N_Z(n)}$, which includes returns from both real targets and clutter (spurious measurements). Here, $\mathbf{z}_{i,n} \in \mathcal{Z}$ is a vector of kinematic measurements (positions in $\mathbb{R}^3$, or $\mathbb{R}^4$ with a Doppler), augmented with an RCS component $R \in \mathbb{R}^+$ for the measured SNR, at time $t_n \in \mathbb{R}$. The assignment vector at time $t_n$ is such that $\mathbf{a}_n(i) = j$ if measurement $i$ is associated with track $j > 0$; $\mathbf{a}_n(i) = 0$ if measurement $i$ is clutter. The inverse mapping $\mathbf{a}_n^{-1}$ maps tracks to measurements: meaning $\mathbf{a}_n^{-1}(\mathbf{a}_n(i)) = i$ if $\mathbf{a}_n(i) \neq 0$; and $\mathbf{a}_n^{-1}(i) = 0 \Leftrightarrow \mathbf{a}_n(j) \neq i$ for all $j$. For example, if $N_T = 4$ and $\mathbf{a} = [2\,0\,0\,1\,4]$ then $N_Z = 5$, $N_c = 2$, and $\mathbf{a}^{-1} = [4\,1\,0\,5]$. Each track is associated with *at most* one measurement, and vice-versa.

In $N$D data association we jointly find the MAP estimate of the association vectors over a sliding window of the last $N - 1$ time steps. We assume we have $N_T(n) \in \mathbb{N}_0$ total tracks as a known parameter: $N_T(n)$ is adjusted over time using various algorithms (see [2, Ch. 3]). In the generative process each track places a probability distribution on the next $N - 1$ measurements, with both kinematic and RCS components. However, if the random RCS $R$ for a measurement is below $R_0$ then it will not be observed. There are $N_c(n) \in \mathbb{N}_0$ clutter measurements from a Poisson process with $\lambda := \mathbb{E}[N_c(n)]$ (often with uniform intensity). The ordering of measurements in $Z_n$ is assumed to be uniformly random. For 3D data association the model joint $p(Z_{n-1:n}, \mathbf{a}_{n-1}, \mathbf{a}_n|Z_{1:n-2})$ is:

$$\prod_{i=1}^{N_T} p_i(\mathbf{z}_{\mathbf{a}_n^{-1}(i),n}, \mathbf{z}_{\mathbf{a}_{n-1}^{-1}(i),n-1}) \times \prod_{i=n-1}^{n} \lambda^{N_c(i)} \exp(-\lambda)/|Z_i|! \prod_{j=1}^{|Z_i|} p_0(\mathbf{z}_{j,i})^{\mathbb{I}\{\mathbf{a}_i(j)=0\}}, \quad (5)$$

where $p_i$ is the probability of the measurement sequence under track $i$; $p_0$ is the clutter distribution. The probability $p_i$ is the product of the RCS component predictions (BOCPD) and the kinematic components (filter); informally, $p_i(\mathbf{z}) = p_i(\text{positions}) \times p_i(\text{RCS})$. If there is a missed detection, i.e. $\mathbf{a}_n^{-1}(i) = 0$, we then use $p_i(\mathbf{z}_{\mathbf{a}_n^{-1}(i),n}) = P(R < R_0)$ under the RCS model for track $i$ with no contribution from positional (kinematic) component. Just as BOCPD allows any black box probabilistic predictor to be used as a UPM, any black box model of measurement sequences can used in (5).

The estimation of association vectors for the 3D case becomes an optimization problem of the form:

$$(\hat{\mathbf{a}}_{n-1}, \hat{\mathbf{a}}_n) = \underset{(\mathbf{a}_{n-1}, \mathbf{a}_n)}{\operatorname{argmax}} \log P(\mathbf{a}_{n-1}, \mathbf{a}_n|Z_{1:n}) = \underset{(\mathbf{a}_{n-1}, \mathbf{a}_n)}{\operatorname{argmax}} \log p(Z_{n-1:n}, \mathbf{a}_{n-1}, \mathbf{a}_n|Z_{1:n-2}), \quad (6)$$

which is effectively optimizing (5) with respect to the assignment vectors. The optimization given in (6) can be cast as a multidimensional assignment (MDA) problem [2], which can be solved efficiently in the 2D case. Higher dimensional assignment problems, however, are NP-hard; approximate, yet typically very accurate, solvers must be used for real-time operation, which is usually required for tracking systems [20].

If a radar scan occurs at each time step and a target is not detected, we assume the SNR has not exceeded the threshold, implying $0 \leq R < R_0$. This is a (left) censored measurement and is treated differently than a missing data point. Censoring is accounted for in Section 2.3.

## 2 Online Variational UPMs

We cover the four technical challenges for implementing non-exponential family UPMs in an efficient and online manner. We drop the index of the data point $i$ when it is clear from context.

### 2.1 Variational Posterior for a Rice Distribution

The Rice distribution has the property that

$$x \sim \mathcal{N}(\nu, \sigma^2), \quad y' \sim \mathcal{N}(0, \sigma^2) \implies R = \sqrt{x^2 + y'^2} \sim \text{Rice}(\nu, \sigma). \tag{7}$$

For simplicity we perform inference using $R^2$, as opposed to $R$, and transform accordingly:

$$x \sim \mathcal{N}(\nu, \sigma^2), \quad R^2 - x^2 \sim \text{Gamma}(\tfrac{1}{2}, \tfrac{\tau}{2}), \quad \tau := 1/\sigma^2 \in \mathbb{R}^+$$
$$\implies p(R^2, x) = p(R^2|x)p(x) = \text{Gamma}(R^2 - x^2|\tfrac{1}{2}, \tfrac{\tau}{2})\mathcal{N}(x|\nu, \sigma^2). \tag{8}$$

The complete likelihood (8) is the product of two exponential family models and is exponential family itself, parameterized with base measure $h$ and partition factor $g$:

$$\eta = [\nu\tau, \, -\tau/2]^\top, \quad \mathcal{S} = [x, \, R^2]^\top, \quad h(R^2, x) = (2\pi\sqrt{R^2 - x^2})^{-1}, \quad g(\nu, \tau) = \tau \exp(-\nu^2\tau/2).$$

By inspection we see that the natural parameters $\eta$ and sufficient statistics $\mathcal{S}$ are the same as a Gaussian with unknown mean and variance. Therefore, we apply the normal-gamma prior on $(\nu, \tau)$ as it is the conjugate prior for the complete data likelihood. This allows us to apply the VB-EM algorithm. We use $y_i := R_i^2$ as the VB observation, not $R_i$ as in (3). In (5), $\mathbf{z}_{\cdot,\cdot}(\text{end})$ is the RCS $R$.

**VB M-Step**  We derive the posterior updates to the parameters given expected sufficient statistics:

$$\bar{x} := \sum_{i=1}^{n} \mathbb{E}[x_i]/n, \quad \mu_n = \frac{\lambda_0 \mu_0 + \sum_i \mathbb{E}[x_i]}{\lambda_0 + n}, \quad \lambda_n = \lambda_0 + n, \quad \alpha_n = \alpha_0 + n, \tag{9}$$

$$\beta_n = \beta_0 + \frac{1}{2}\sum_{i=1}^{n}(\mathbb{E}[x_i] - \bar{x})^2 + \frac{1}{2}\frac{n\lambda_0}{\lambda_0 + n}(\bar{x} - \mu_0)^2 + \frac{1}{2}\sum_{i=1}^{n}R_i^2 - \mathbb{E}[x_i]^2. \tag{10}$$

This is the same as an observation from a Gaussian and a gamma that share a (inverse) scale $\tau$.

**VB E-Step**  We then must find both expected sufficient statistics $\bar{\mathcal{S}}$. The expectation $\mathbb{E}[R_i^2|R_i^2] = R_i^2$ trivially; leaving $\mathbb{E}[x_i|R_i^2]$. Recall that the joint on $(x, y')$ is a bivariate normal; if we constrain the radius to $R$, the angle $\omega$ will be distributed by a *von Mises* (VM) distribution. Therefore,

$$\omega := \arccos(x/R) \sim \text{VM}(0, \kappa), \quad \kappa = R\,\mathbb{E}[\nu\tau] \implies \mathbb{E}[x] = R\,\mathbb{E}[\cos\omega] = RI_1(\kappa)/I_0(\kappa), \tag{11}$$

where computing $\kappa$ constitutes the VB E-step and we have used the trigonometric moment on $\omega$ [18]. This completes the computations required to do the VB updates on the Rice posterior.

**Variational Lower Bound**  For completeness, and to assess convergence, we derive the VB lower bound $\mathcal{L}(q)$. Using the standard formula [4] for $\mathcal{L}(q) = \mathbb{E}_q[\log p(\mathbf{y}_{1:n}, \mathbf{x}_{1:n}, \theta)] + \text{H}[q]$ we get:

$$\sum_{i=1}^{n} \mathbb{E}[\log \tau/2] - \tfrac{1}{2}\mathbb{E}[\tau]R_i^2 + (\mathbb{E}[\nu\tau] - \kappa_i/R_i)\mathbb{E}[x_i] - \tfrac{1}{2}\mathbb{E}[\nu^2\tau] + \log I_0(\kappa_i) - \text{KL}(q\|p), \tag{12}$$

where $p$ in the KL is the prior on $(\nu, \tau)$ which is easy to compute as $q$ and $p$ are both normal-gamma. Equivalently, (12) can be optimized directly instead of using the VB-EM updates.

### 2.2 Online Variational Inference

In Section 2.1 we derived an efficient way to compute the variational posterior for a Rice distribution for a fixed data set. However, as is apparent from (1) we need online predictions from the UPM; we must be able to update the posterior one data point at a time. When the UPM is exponential family and we can compute the posterior exactly, we merely use the posterior from the previous step as the prior. However, since we are only computing a variational approximation to the posterior, using the previous posterior as the prior does not give the exact same answer as re-computing the posterior from batch. This gives two obvious options: 1) recompute the posterior from batch every update at $\mathcal{O}(n)$ cost or 2) use the previous posterior as the prior at $\mathcal{O}(1)$ cost and reduced accuracy.

The difference between the options is encapsulated by looking at the expected sufficient statistics: $\bar{S} = \sum_{i=1}^{n} \mathbb{E}_{q(x_i|\mathbf{y}_{1:n})}[\mathcal{S}(x_i, y_i)]$. Naive online updating uses old expected sufficient statistics whose posterior effectively uses $\bar{S} = \sum_{i=1}^{n} \mathbb{E}_{q(x_i|\mathbf{y}_{1:i})}[\mathcal{S}(x_i, y_i)]$. We get the best of both worlds if we adjust those estimates over time. We in fact can do this if we project the expected sufficient statistics into a "feature space" in terms of the expected natural parameters. For some function $f$,

$$q(x_i) = p(x_i|y_i, \eta = \bar{\eta}) \implies \mathbb{E}_{q(x_i|\mathbf{y}_{1:n})}[\mathcal{S}(x_i, y_i)] = f(y_i, \bar{\eta}). \qquad (13)$$

If $f$ is piecewise continuous then we can represent it with an inner product [8, Sec. 2.1.6]

$$f(y_i, \bar{\eta}) = \phi(\bar{\eta})^\top \psi(y_i) \implies \bar{S} = \sum_{i=1}^{n} \phi(\bar{\eta})^\top \psi(y_i) = \phi(\bar{\eta})^\top \sum_{i=1}^{n} \psi(y_i), \qquad (14)$$

where an infinite dimensional $\phi$ and $\psi$ may be required for exact representation, but can be approximated by a finite inner product. In the Rice distribution case we use (11)

$$f(y_i, \bar{\eta}) = \mathbb{E}[x_i] = R_i I'(R_i \mathbb{E}[\nu\tau]) = R_i I'((R_i/\mu_0) \mu_0 \mathbb{E}[\nu\tau]), \quad I'(\cdot) := I_1(\cdot)/I_0(\cdot), \qquad (15)$$

where recall that $y_i = R_i^2$ and $\bar{\eta}_1 = \mathbb{E}[\nu\tau]$. We can easily represent $f$ with an inner product if we can represent $I'$ as an inner product: $I'(uv) = \phi(u)^\top \psi(v)$. We use unitless $\phi_i(u) = I'(c_i u)$ with $\mathbf{c}_{1:G}$ as a log-linear grid from $10^{-2}$ to $10^3$ and $G = 50$. We use a lookup table for $\psi(v)$ that was trained to match $I'$ using non-negative least squares, which left us with a sparse lookup table. Online updating for VB posteriors was also developed in [24; 13]. These methods involved introducing forgetting factors to forget the contributions from old data points that might be detrimental to accuracy. Since the VB predictions are "embedded" in a change point method, they are automatically phased out if the posterior predictions become inaccurate making the forgetting factors unnecessary.

## 2.3 Censored Data

As mentioned in Section 1.3, we must handle censored RCS observations during a missed detection. In the VB-EM framework we merely have to compute the expected sufficient statistics given the censored measurement: $\mathbb{E}[\mathcal{S}|R < R_0]$. The expected sufficient statistic from (11) is now:

$$\mathbb{E}[x|R < R_0] = \int_0^{R_0} \mathbb{E}[x|R]p(R)dR \,/\, \text{Rice}_{\text{CDF}}(R_0|\nu, \tau) = \nu(1 - Q_2(\tfrac{\nu}{\sigma}, \tfrac{R_0}{\sigma}))/(1 - Q_1(\tfrac{\nu}{\sigma}, \tfrac{R_0}{\sigma})),$$

where $Q_M$ is the Marcum $Q$ function [17] of order $M$. Similar updates for $\mathbb{E}[\mathcal{S}|R < R_0]$ are possible for exponential or gamma UPMs, but are not shown as they are relatively easy to derive.

## 2.4 Variational Run Length Posteriors: Predictive Log Likelihoods

Both updating the BOCPD run length posterior (1) and finding the marginal predictive log likelihood of the next point (2) require calculating the UPM's posterior predictive log likelihood $\log p(y_{n+1}|r_n, \mathbf{y}_{(r)})$. The marginal posterior predictive from (2) is used in data association (6) and benchmarking BOCPD against other methods. However, the exact posterior predictive distribution obtained by integrating the Rice likelihood against the VB posterior is difficult to compute.

We can break the BOCPD update (1) into a time and measurement update. The measurement update corresponds to a Bayesian model comparison (BMC) calculation with prior $p(r_n|\mathbf{y}_{1:n})$:

$$p(r_n|\mathbf{y}_{1:n+1}) \propto p(y_{n+1}|r_n, \mathbf{y}_{(r)})p(r_n|\mathbf{y}_{1:n}). \qquad (16)$$

Using the BMC results in Bishop [4, Sec. 10.1.4] we find a variational posterior on the run length by using the variational lower bound for each run length $\mathcal{L}_i(q) \leq \log p(y_{n+1}|r_n = i, \mathbf{y}_{(r)})$, calculated using (12), as a proxy for the exact UPM posterior predictive in (16). This gives the exact VB posterior if the approximating family $\mathcal{Q}$ is of the form:

$$q(r_n, \theta, x) = q_{\text{UPM}}(\theta, x|r_n)q(r_n) \implies q(r_n = i) = \exp(\mathcal{L}_i(q))p(r_n=i|\mathbf{y}_{1:n})/\exp(\mathcal{L}(q)), \qquad (17)$$

where $q_{\text{UPM}}$ contains whatever constraints we used to compute $\mathcal{L}_i(q)$. The normalizer on $q(r_n)$ serves as a joint VB lower bound: $\mathcal{L}(q) = \log \sum_i \exp(\mathcal{L}_i(q))p(r_n=i|\mathbf{y}_{1:n}) \leq \log p(y_{n+1}|\mathbf{y}_{1:n})$. Note that the conditional factorization is different than the typical independence constraint on $q$.

Furthermore, we derive the estimation of the assignment vectors $\mathbf{a}$ in (6) as a VB routine. We use a similar conditional constraint on the latent BOCPD variables given the assignment and constrain the assignment posterior to be a point mass. In the 2D assignment case, for example,

$$q(\mathbf{a}_n, \mathcal{X}_{1:N_T}) = q(\mathcal{X}_{1:N_T}|\mathbf{a}_n)q(\mathbf{a}_n) = q(\mathcal{X}_{1:N_T}|\mathbf{a}_n)\mathbb{I}\{\mathbf{a}_n = \hat{\mathbf{a}}_n\}, \qquad (18)$$

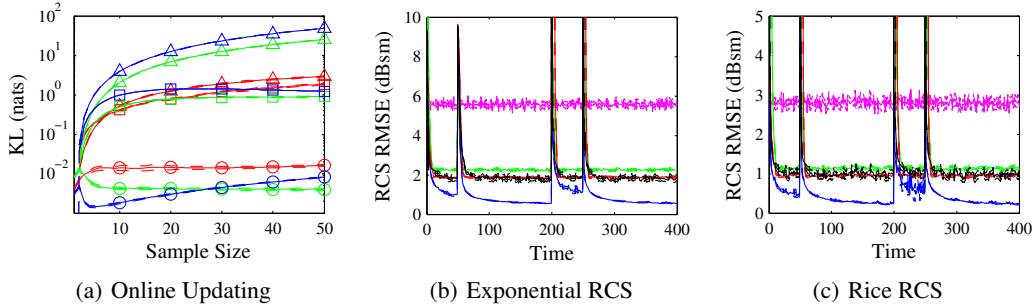

<figure>
(a) Online Updating      (b) Exponential RCS      (c) Rice RCS
</figure>

Figure 2: **Left:** KL from naive updating ($\triangle$), Sato's method [24] ($\square$), and improved online VB ($\circ$) to the batch VB posterior vs. sample size $n$; using a standard normal-gamma prior. Each curve represents a true $\nu$ in the generating Rice distribution: $\nu = 3.16$ (red), $\nu = 10.0$ (green), $\nu = 31.6$ (blue) and $\tau = 1$. **Middle:** The RMSE (dB scale) of the estimate on the mean RCS distribution $\mathbb{E}[R_n]$ is plotted for an exponential RCS model. The curves are BOCPD (blue), IMM (black), identity (magenta), $\alpha$-filter (green), and median filter (red). **Right:** Same as the middle but for the Rice RCS case. The dashed lines are 95% confidence intervals.

where each track's $\mathcal{X}_i$ represents all the latent variables used to compute the variational lower bound on $\log p(\mathbf{z}_{j,n}|\mathbf{a}_n(j)\!=\!i)$. In the BOCPD case, $\mathcal{X}_i := \{r_n, x, \theta\}$. The resulting VB fixed point equations find the posterior on the latent variables $\mathcal{X}_i$ by taking $\hat{\mathbf{a}}_n$ as the true assignment and solving the VB problem of (17); the assignment $\hat{\mathbf{a}}_n$ is found by using (6) and taking the joint BOCPD lower bound $\mathcal{L}(q)$ as a proxy for the BOCPD predictive log likelihood component of $\log p_i$ in (5).

## 3 Results

### 3.1 Improved Online Solution

We first demonstrate the accuracy of the online VB approximation (Section 2.2) on a Rice estimation example; here, we only test the VB posterior as no change point detection is applied. Figure 2(a) compares naive online updating, Sato's method [24], and our improved online updating in KL(online∥batch) of the posteriors for three different true parameters $\nu$ as sample size $n$ increases. The performance curves are the KL divergence between these online approximations to the posterior and the batch VB solution (i.e. restarting VB from "scratch" every new data point) vs sample size. The error for our method stays around a modest $10^{-2}$ nats while naive updating incurs large errors of 1 to 50 nats [19, Ch. 4]. Sato's method tends to settle in around a 1 nat approximation error. The recommended annealing schedule, i.e. forgetting factors, in [24] performed worse than naive updating. We did a grid search over annealing exponents and show the results for the best performing schedule of $n^{-0.52}$. By contrast, our method does not require the tuning of an annealing schedule.

### 3.2 RCS Estimation Benchmarking

We now compare BOCPD with other methods for RCS estimation. We use the same experimental example as Slocumb and Klusman III [25], which uses an augmented interacting multiple model (IMM) based method for estimating the RCS; we also compare against the same $\alpha$-filter and median filter used in [25]. As a reference point, we also consider the "identity filter" which is merely an unbiased filter that uses *only* $y_n$ to estimate the mean RCS $\mathbb{E}[R_n]$ at time step $n$. We extend this example to look at Rice RCS in addition to the exponential RCS case. The bias correction constants in the IMM were adjusted for the Rice distribution case as per [25, Sec. 3.4].

The results on exponential distributions used in [25] and the Rice distribution case are shown in Figures 2(b) and 2(c). The IMM used in [25] was hard-coded to expect jumps in the SNR of multiples of $\pm 10$ dB, which is exactly what is presented in the example (a sequence of 20, 10, 30, and 10 dB). In [25] the authors mention that the IMM reaches an RMSE "floor" at 2 dB, yet BOCPD continues to drop as low as 0.56 dB. The RMSE from BOCPD does not spike nearly as high as the other methods upon a change in $\mathbb{E}[R_n]$. The $\alpha$-filter and median filter appear worse than both the IMM and BOCPD. The RMSE and confidence intervals are calculated from 5000 runs of the experiment.

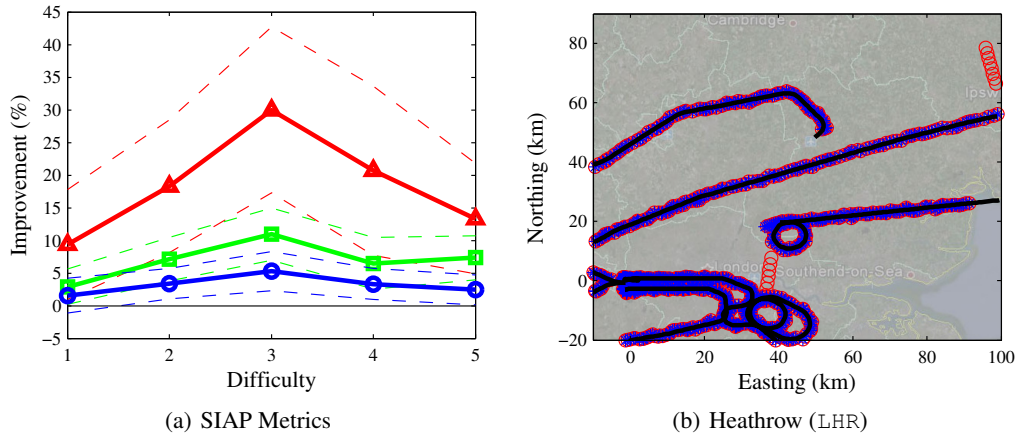

| (a) SIAP Metrics | (b) Heathrow (LHR) |

Figure 3: **Left:** Average relative improvements (%) for SIAP metrics: position accuracy (red △), velocity accuracy (green □), and spurious tracks (blue ○) across difficulty levels. **Right:** LHR: true trajectories shown as black lines (−), estimates using a BOCPD RCS model for association shown as blue stars (∗), and the standard tracker as red circles (○). The standard tracker has spurious tracks over east London and near Ipswich.

Background map data: Google Earth (TerraMetrics, Data SIO, NOAA, U.S. Navy, NGA, GEBCO, Europa Technologies)

### 3.3 Flightradar24 Tracking Problem

Finally, we used real flight trajectories from flightradar24 and plugged them into our 3D tracking algorithm. We compare tracking performance between using our BOCPD model and the relatively standard constant probability of detection (no RCS) [2, Sec. 3.5] setup. We use the single integrated air picture (SIAP) metrics [6] to demonstrate the improved performance of the tracking. The SIAP metrics are a standard set of metrics used to compare tracking systems. We broke the data into 30 regions during a one hour period (in Sept. 2012) sampled every 5 s, each within a 200 km by 200 km area centered around the world's 30 busiest airports [22]. Commercial airport traffic is typically very orderly and does not allow aircraft to fly close to one another or cross paths. Feature-aided tracking is most necessary in scenarios with a more chaotic air situation. Therefore, we took random subsets of 10 flight paths and randomly shifted their start time to allow for scenarios of greater interest.

The resulting SIAP metric improvements are shown in Figure 3(a) where we look at performance by a difficulty metric: the number of times in a scenario any two aircraft come within ∼400 m of each other. The biggest improvements are seen for difficulties above three where positional accuracy increases by 30%. Significant improvements are also seen for velocity accuracy (11%) and the frequency of *spurious* tracks (6%). Significant performance gains are seen at all difficulty levels considered. The larger improvements at level three over level five are possibly due to some level five scenarios that are not resolvable simply through more sophisticated models. We demonstrate how our RCS methods prevent the creation of spurious tracks around London Heathrow in Figure 3(b).

## 4 Conclusions

We have demonstrated that it is possible to use sophisticated and recent developments in machine learning such as BOCPD, and use the modern inference method of VB, to produce demonstrable improvements in the much more mature field of radar tracking. We first closed a "hole" in the literature in Section 2.1 by deriving variational inference on the parameters of a Rice distribution, with its inherent applicability to radar tracking. In Sections 2.2 and 2.4 we showed that it is possible to use these variational UPMs for non-exponential family models in BOCPD without sacrificing its modular or online nature. The improvements in online VB are extendable to UPMs besides a Rice distribution and more generally beyond change point detection. We can use the variational lower bound from the UPM and obtain a principled variational approximation to the run length posterior. Furthermore, we cast the estimation of the assignment vectors themselves as a VB problem, which is in large contrast to the tracking literature. More algorithms from the tracking literature can possibly be cast in various machine learning frameworks, such as VB, and improved upon from there.

## Footnotes

[1] The shape $\nu$ is usually assumed to be positive ($\in \mathbb{R}^+$); however, there is nothing wrong with using a negative $\nu$ as $\mathrm{Rice}(x|\nu, \sigma) = \mathrm{Rice}(x|-\nu, \sigma)$. It also allows for use of a normal-gamma prior.

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
