[Reviews · NeurIPS 2013]

Submitted by Assigned_Reviewer_4

This is an interesting and novel paper about using variational
inference in Bayesian online change point detection. It seemed to me
that the experimental results are interesting and thorough, though I
am not in the field of radar tracking.

A problem with the paper is that I found the exposition hard to
follow---a more holistic view of the algorithm is needed to make the
paper clearer. (That is, after reading the paper, I'm not sure I could
implement it.) Perhaps you can include pseudocode with pointers to
various equations.

Comments:

p3: Intuitions about the Rice distribution would be nice. When and
why is it used in tracking?

p3: Nitpick: Variational Bayes finds a *local* optimum.

p4: Variational inference is easiest in the settings you describe,
those that are conjugate-exponential with latent variables. But see
recent work like [1] and [2] which seek to expand on this.

p7: I was intrigued by the comment "improved online updating in terms
of KL to batch" but could not unpack it---I don't think that turn of
phrase appears earlier to explain an idea in the algorithm. Here again
a holistic summary of the algorithm would help make the paper clearer.

p8: In the conclusions, "closing a hole" is a little strong. You
needed a variational approximation to the Rice so you developed it.
This is straightforward because it is a conjguate-exponential model.
(This is not to take away from the contribution---you make a good case
via an interesting application for the Rice being a distribution for
which we want a variational approximation.)

Also, I found referring to "non-exponential family distributions"
confusing. Yes, the Rice is not in the exponential family but, because
I read many papers about variational inference, I read the expression
to mean latent variable models that are not in the
conjugate-exponential family, which the Rice is in. (No need to change
if you feel otherwise.)
Summary: This is an interesting paper. I am not in the field of radar tracking, but it seemed like a novel and important application of variational inference.

Submitted by Assigned_Reviewer_5

The paper presents a variational Bayes approach for online change-point detection in Rice distributed signal, accounting for the censoring of some observations, in the perspective of radar tracking. As far as I understand, the goal is to associate radar signals with real objects, which sounds like a unsupervised classification problem. Change point detection come on the top of that, due to changes in the data acquisition process.
One of the contribution is the variational posterior for a Rice distribution which does no belong to the exponential family, but that can be re-parametrized as a combination of exponential family distribution involving one unobserved variable $x$. The trick is nice, but then resumes to a classical variational Bayes approximation. A series of calculations derived in the paper resort from the same family, used in a relevant manner, but not really new.
I found this paper very hard to read for several reasons. The authors define more than 10 different acronyms. The model is never clearly stated so conditional dependences or independences have to be guessed, e.g. from equation (1). The connection between sections 1.2 and 1.3 is a bit hard to find. It is not clear to me if the Z (resp. a) of section 1.3 plays the role of the observed y (resp. unobserved x) of section 1.2. Also the connexion between classification (data association) and change point detection is still a bit obscure to me.
Summary: Maybe an interesting paper, but very hard to read.

Submitted by Assigned_Reviewer_6

This paper develops a Bayesian approach to change point models using a variational approximation. The basic approach is not new, but the distribution used here (Rice) and the approximation needed for handling it are unpublished. In particular, the way they modify the Rice to facilitate VB, while not deep, is technically clever and might find uses in other models. The approach to online learning is an interesting twist on Sato's work, but I miss evidence that it is necessary or better. There's probably a way to make this section clearer. I do not have sufficient domain expertise to judge the significance of the results on the 2 datasets (Slocumb and Klusman, flightradar24) but they seem impressive.

Quality: the paper is technically solid.
Clarity: the paper is well organized but not always easy to follow.
Originality: the approach taken by the paper is not in itself novel, but the application domain is relatively rare for this type of approach, and the paper adds technical novelties.
Significance: the problem addressed is difficult and the results seem to improve on previous approaches.
Summary: This paper addresses the problem of multibody tracking in a radar context, which is quite hard. It is technically sound, and the approach contains some interesting technical novelties. Results seem convincing.

Author Feedback

Author rebuttal: We thank the reviewers for their useful comments.

Reviewer 4 motivates the use of a few extra citations. The use of a Rice distribution is motivated by the physics of electromagnetic reflections; a more modern citation than [25] (i.e. Swerling 1954) may be helpful for those interested. Sufficient detail to review the background of coding an entire tracking system is difficult to cover in the background section of a NIPS paper but augmenting the current citations of [2] and [20] might be helpful.

Reviewer 4 observes that the case of VB for a Rice noise model is simplified by the fact that it belongs to a class of models that are conjugate exponential family after being augmented with latent variables. However, as the reviewers note, this is a much larger class than exponential family since it includes Rice, Student-t, and mixture of Gaussians, just to name a few examples. Additionally, the VB bounds of Section 2.4 hold even if there is no clear way to augment the model to make it conjugate exponential.

Reviewers 4 and 6 would probably like if we added a few more sentences clarifying the performance benefits of the explained online method over Sato's method shown in Figure 2(a). To elaborate what is stated in the paper: The performance curves are the KL divergence between various online approximations to the posterior and the batch VB solution (i.e. redoing VB from scratch every new data point) vs sample size. Sato's online approximation to the posterior tends to have a KL divergence to the batch solution of around 1 nat, where as our approximation has an error of about 0.01 nats, which is a 100x improvement. On an absolute scale, 0.01 nats is typically considered a small approximation error where as 1 nat is a large one (see Kass and Raftery [1995] and Murray [2007, Ch. 4]).

Reviewer 5 appears to have some questions about the connections between different sub-sections of the background section. In our application, the radar cross section (RCS) measurement is used as the observed variable in VB (y in section 1.2). In practice, RCS will typically be one of the components of the z vector along with kinematic measurements (positions and perhaps velocity). So, the reviewer is effectively correct in that the Z measurements of Section 1.3 (background on tracking) are used as the observed y of Section 1.2 (background on VB). However, z will also contain measurements from other sensors that are ignored in BOCPD and VB.

Reviewer 5 would also like some extra explanation on the conditional independencies or what is sometimes referred to as the "probability of everything" or the joint of the model. We do provide the full joint for the tracking problem in (5) as we determined that would be most useful in explaining tracking to a machine learning audience; writing the full joint is not as common in the tracking literature. For space reasons we did not state the full joint of the BOCPD model (and its implied conditional independencies) as it is already present in the cited background on BOCPD.